# Managing Dry Eye Disease with Novel Medications: Mechanism, Study Validity, Safety, Efficacy, and Practical Application

**DOI:** 10.3390/pharmacy12010019

**Published:** 2024-01-23

**Authors:** Jason C. Wong, Aselle Barak

**Affiliations:** 1Pharmacy Practice and Administration, Western University of Health Sciences, Pomona, CA 91766, USA; 2College of Pharmacy, Western University of Health Sciences, Pomona, CA 91766, USA; aselle.barak@westernu.edu

**Keywords:** dry eye disease, lifitegrast, cyclosporine, loteprednol etabonate, varenicline nasal spray, perfluorohexyloctane

## Abstract

Dry eye disease (DED) is a common condition that affects mainly older individuals and women. It is characterized by reduced tear production and increased tear evaporation. Symptoms include burning, irritation, tearing, and blurry vision. This paper reviews key trials of various new DED treatments, including their mechanism of action, study outcomes, safety, and efficacy. The paper also includes a critical assessment of the trial’s validity and potential pharmacy applications of these new treatments. The literature search was conducted through PubMed, the Cochrane Central Register of Controlled Trials, and Google Scholar. The keywords “Dry Eye Disease”, “lifitegrast”, “cyclosporine”, “loteprednol etabonate”, “varenicline nasal spray”, and “perfluorohexyloctane” were used to identify these medications’ landmark trials. The articles deemed these medications safe and efficacious, with minimal side effects. Our randomized controlled trial validity comparison found the trials robust with predominantly low bias. Cyclosporine and loteprednol are effective when artificial tears fail, while perfluorohexyloctane reduces tear film evaporation and is preservative-free. Varenicline offers drug delivery via the nasal route and is appropriate for contact lens users. In conclusion, these FDA-approved novel medications exhibit safety and efficacy in managing DED. Further research is needed on long-term outcomes, efficacy, and side-effect comparisons, and combination therapy benefits.

## 1. Introduction

Dry eye disease (DED), also known as keratoconjunctivitis sicca (KCS), is an eye condition caused by an imbalance between tear production and evaporation. This leads to inflammation of the eye surface, resulting in eye irritation and potential corneal damage. DED can cause dysregulated tear secretion and reduced tear production, leading to unwanted symptoms for patients [1].

In accordance with the International Dry Eye Workshop II (DEWS II)’s classification, DED is categorized into two major types: aqueous deficient dry eye (ADDE) and evaporative dry eye (EDE) disease. ADDE occurs due to insufficient lacrimal secretion, whereas EDE results from excessive water loss. However, both conditions may coexist, producing a mixed type of DED [2].

ADDE can be categorized into Sjögren syndrome (SS) dry eye or Non-Sjögren (non-SS) dry eye. Individuals with SS dry eye exhibit coexisting connective tissue disease or xerostomia, classifying them as primary or secondary SS. Secondary SS occurs in those with SS-like symptoms that have progressed within a diagnosed CTD, such as psoriatic arthritis, systemic sclerosis, rheumatoid arthritis, and systemic lupus erythematosus [3].

EDE can be intrinsic or extrinsic. Intrinsic EDE is caused by meibomian gland disorder, slow blink rate, and isotretinoin. Extrinsic EDE is caused by factors such as drug preservatives, vitamin A deficiency, contact lens usage, and allergic conjunctivitis [4]. Refer to Figure 1 for the DEWS II classification of DED.

Common risk factors for DED include vitamin A deficiency, the use of contact lenses, medication usage (diuretics, antihistamines, isotretinoin, etc.), a sedentary lifestyle, metabolic syndrome (characterized by an increase in blood pressure, blood glucose levels, and abnormal cholesterol levels), and anxiety/depression [5]. The administration of ophthalmic medications may inadvertently harm the tear film, exacerbate ocular inflammation, and increase tear evaporation due to potential administration errors. For example, the mishandling of medication can occur when an excessive amount is instilled in the affected eye or when multiple topical ophthalmics are used concurrently without proper time separation between each use, if applicable. Ophthalmic products with preservatives are another factor that can contribute to the progression of DED. Toxic preservatives can potentially inflict additional damage on the ocular tissues [4]. Benzalkonium chloride is a well-known preservative that causes eye irritation and ocular damage. Eliminating factors such as medication and contact lens use can significantly improve symptoms. However, if left untreated, DED may progress and lead to worsening symptoms. In cases where DED advanced from mild to severe, patients may develop irreversible inflammatory conditions, including vision loss, corneal scarring, thinning, or neovascularization [1]. 

DED is most commonly identified in women and individuals aged 50 and above. It is estimated that 4.88 million individuals over 50 years of age in the U.S. experience DED, with a projected rise in prevalence among young adults aged 18–34 [5]. Older adults are at a higher risk for DED due to reduced tear production caused by various factors such as the use of medications, hormonal changes, and autoimmune diseases like SS and rheumatoid arthritis [6,7]. The surge in DED among young adults can be attributed to the widespread use of electronic devices [8]. As technology advances, an annual increase of over 10% in DED is anticipated in young adults [9]. DED affects quality of life and poses an economic burden due to reduced work productivity [10].

While DED is not curable, its symptoms can be managed. The 2018 American Academy of Ophthalmology (AAO) guidelines recommend four steps to manage and treat DED. Step 1 includes modifying the environment, diet, and lid hygiene, and using ocular lubricants like artificial tears. Step 2 suggests using non-preserved lubricants, tear conservation, and prescription drugs. Step 3 may include eye drops, oral secretagogues, and contact lenses. Step 4 includes surgical punctal occlusion, amniotic membrane grafts, long-term topical corticosteroid use, and other surgical approaches [1]. Refer to Figure 2 for the 2018 AAO treatment algorithm for DED.

The management of symptoms of DED often involves lifestyle modifications that, when taken with proper precautions, can provide relief. Environmental changes, such as avoiding dust, smoke, and wind exposure, have proven to be beneficial. Incorporating aerobic exercise may stimulate tear secretion. For contact lens wearers, reducing usage or discontinuing their use is advisable. Furthermore, integrating low-glycemic-index foods like vegetables and whole grains may also help alleviate dry eye symptoms [11].

Adding essential fatty acids (EFAs) to one’s diet can reduce inflammation and alleviate dry eye symptoms [12]. Omega-3 EFAs can slow tear evaporation rates and increase tear secretion [13]. The omega-3/omega-6 EFA ratio is critical in controlling inflammation, but Western diets typically have a ratio of 15:1, far from the ideal ratio of 4:1 [14].

The pharmaceutical industry has recently introduced several DED medications that have received FDA approval. Nonetheless, research into these medications’ safety, effectiveness, similarities, and differences is currently inadequate. This review aims to review the key trials of the various new DED treatments, focusing on their mechanism of action, study outcomes, safety, and efficacy. A critical assessment of the study’s trial validity and a brief discussion of the potential pharmacy applications of these new treatments are included.

## 2. Materials and Methods

A PubMed search was performed using keywords “dry eye disease”, “lifitegrast”, “cyclosporine”, “loteprednol etabonate”, “varenicline”, and “perfluorohexyloctane” to find the landmark trials that resulted in the FDA approval of these five novel drugs. Additional search filters included Randomized Controlled Trial (article type), 1 January 2015–5 June 2023 (publication date), Humans (species), and English (article language). The PubMed search generated 920 results, with seven being of importance to the approval of Lifitegrast (5.0%), Cyclosporine (0.09%), Loteprednol etabonate (0.25%), Varenicline nasal spray, and Perfluorohexyloctane. See Figure 3 for details.

To evaluate the validity of the seven randomized control trials (RCT), a critical appraisal tool (FRISBE mnemonic) was used. The mnemonic stands for “follow up”, “randomization”, “intent-to-treat analysis”, similar baseline characteristics”, “blinding”, and “equal treatment”. 

## 3. Results

### 3.1. Lifitegrast

Lifitegrast (Xiidra) ophthalmic solution received FDA approval in 2016. It targets lymphocyte function-associated antigen 1 (LFA-1) to address inflammation associated with DED. Lifitegrast reduces T-cell activation, inhibiting the interaction between the intercellular adhesion molecule (ICAM-1) and integrin LFA antigen 1 [15]. Ultimately, this inhibition mitigates inflammation and represents a promising target for ophthalmic agents to prevent T-cell-mediated inflammation [16]. 

In a phase 3 randomized, multi-centered, placebo-controlled, double-masked clinical trial, the safety and efficacy of Lifitegrast 5% was observed in seven hundred and eighteen participants. Subjects were randomized into the Lifitegrast group (n = 358) or placebo group (n = 360). The primary endpoint evaluated the eye dryness score (EDS) and inferior corneal fluorescein staining (iCFS) score, with secondary endpoints including total corneal fluorescein staining (tCFS) score, ocular discomfort, and eye discomfort. Ocular treatment-emergent AEs (TEAEs) were also observed in this study. The results showed a statistically significant improvement in EDS (*p* < 0.0001) with Lifitegrast. However, there was no difference in iCFS (*p* = 0.6186). Secondary endpoint results revealed clinically significant differences in eye discomfort (*p* < 0.0001) and ocular discomfort (*p* = 0.0005) amongst the lifitegrast group. However, the tCFS score did not show any difference. Ocular TEAEs were more commonly seen in the lifitegrast group compared to placebo: 33.7% and 16.4%, respectively [17]. Overall, the primary and secondary endpoints assessment concluded that using lifitegrast 5% twice daily significantly improves DED symptoms.

The randomized controlled trial validity assessment for lifitegrast is summarized in Table 1. 

### 3.2. Cyclosporine

Cyclosporine 0.09% (Cequa) is a preservative-free (PF) ophthalmic solution that was approved by the FDA in 2018 [16]. As a calcineurin inhibitor, this drug evinces immunomodulatory characteristics by hindering the activation, penetration, and discharge of inflammatory cytokines from T cells. This mechanism is the primary way by which it improves symptoms associated with DED. Cyclosporine 0.09% functions by binding to cyclophilin and forming a complex to prevent calcineurin-mediated dephosphorylation. It utilizes NCELL technology to enhance the delivery of active ingredients to the ocular surface. This technology involves nanomicelles composed of polymers that encapsulate the cyclosporine molecules, providing a protective layer and minimizing the disintegration of cyclosporine. The nanomicelles are formulated with size reduction and amphipathic properties, facilitating easy entry into corneal and conjunctival cells. Upon entry, a high concentration of cyclosporine is released, acting on ocular cells to provide conjunctival staining and increase tear production. Due to its specific drug design, cyclosporine 0.09% offers 3 times more penetration and 1.6 times more conjunctival absorption to the ocular surface compared to lifitegrast 5.0% and cyclosporine 0.05%. The effect of cyclosporine 0.09% can occur within a few weeks of treatment [18].

In a phase 3 multicenter, randomized, vehicle-controlled, double-masked clinical study, seven hundred and forty-four patients were randomized into the OTX-101 (cyclosporine 0.09%) group (n = 371) or the vehicle group (n = 373) The primary endpoint was ≥ 10 mm changes in Schirmer test scores (STS) with results showing a statistically significant difference in the OTX-101 group compared to the vehicle group (*p* < 0.001). Additionally, safety evaluations were completed, and the study demonstrated that OTX-101 was well-tolerated with mild TEAEs [19]. 

The randomized controlled trial validity assessment for Cyclosporine 0.09% is summarized in Table 2.

### 3.3. Loteprednol Etabonate

Loteprednol etabonate 0.25% (Eysuvis) is an ophthalmic corticosteroid suspension that gained approval in October 2020 from the FDA for managing short-term DED symptoms. It was developed using mucus-penetrating particle (MPP) technology, a distinctive approach designed to facilitate drug delivery to the mucosal surfaces. In developing loteprednol etabonate 0.25%, polymeric nanoparticles were coated with a high-density and low molecular weight polymer. The polymer coating reduces the binding affinity of particles to mucins, thereby enabling a significant improvement in drug delivery to mucosal surfaces. The recommended dosage is instilling one to two drops into the affected eye four times daily. However, because it is an ophthalmic steroid, its use is limited to up to two weeks of treatment [20].

Loteprednol etabonate 0.25%’s safety and efficacy profile were evaluated in one phase 2 trial and three phase 3 trials (STRIDE 1, 2 and 3). All trials were double-masked, randomized, multicentered, vehicle-controlled studies with approximately 2800 participants. The primary endpoint for STRIDE 1 and 2 studies was the change in bulbar conjunctival hyperemia and ocular discomfort (patient-reported). For STRIDE 3, the primary endpoint was a change in ocular discomfort. The secondary endpoint for STRIDE 3 was conjunctival hyperemia. All three phase 3 trials showed a more noticeable reduction in conjunctival hyperemia amongst the loteprednol etabonate 0.25% (KPI-121) group compared to the vehicle group. Ocular discomfort was significantly reduced amongst the KPI-121 group in STRIDE 1 and 3. In STRIDE 2 and 3, there was a clinically significant change in tCFS with KPI-121 compared to the vehicle group. The most frequently reported AE was instillation site pain [21].

One study assessed the safety of KPI-121 in a pooled analysis of the four aforementioned clinical trials. The most common ocular AE was mild instillation site pain (<5.5%). TEAEs were also measured and reported in 1.0% of subjects in both groups. Furthermore, this study demonstrated KPI-121 to be safe and well-tolerated [22].

The randomized controlled trial validity assessment for loteprednol etabonate is summarized in Table 3.

### 3.4. Perfluorohexyloctane

Perfluorohexyloctane (Miebo) is a topical ophthalmic solution most recently approved in May 2023 by the FDA to treat symptoms associated with DED [23]. Perfluorohexyloctane is an anhydrous, semi-fluorinated alkane that directly targets tear evaporation. While the precise way in which the drug works is not yet entirely known, it creates a single layer of molecules on the surface where the air and the liquid components of the eye’s tear film meet. This monolayer minimizes tear evaporation, contributing to its therapeutic effect in managing DED [24]. It is suggested to instill one drop into the affected eye four times a day [23].

In a phase 3 randomized, multicenter, double-masked, saline-controlled study (GOBI study), five hundred and ninety-seven patients were randomized into the Perfluorohexyloctane (NOV03) group (n = 303) or hypotonic saline group (n = 294). The primary endpoint centered on the change in tCFS score and EDS, while secondary endpoints included changes in EDS and tCFS score, burning or stinging score, and central corneal fluorescein staining (cCFS) score. It was observed that there was a statistically significant improvement in tCFS and EDS (*p* < 0.001) as well as all secondary endpoints (*p* < 0.01). Mild ocular AEs were reported by 9.6% of NOV03 group subjects and 7.5% of saline group subjects. The most common AEs observed in the NOV03 group were instillation site pain, blurred vision, and eye discharge [25].

The randomized controlled trial validity assessment for perfluorohexyloctane, GOBI Study, is summarized in Table 4.

The MOJAVE study was a phase 3 randomized, multicenter, double-masked, saline-controlled trial that analyzed the efficacy and safety of Perfluorohexlyoctane. Five hundred and ninety-seven patients were randomized into the Perfluorohexlyoctane (NOV03) group (n = 303) or saline group (n = 294). The primary endpoint included a change in the tCFS score and EDS. The secondary endpoints included a change in tCFS, EDS, cCFS, and burning or stinging score. The safety of the treatment was assessed by measuring ocular adverse events (AEs). The NOV03 group showed a statistically significant improvement in tCFS and EDS compared to the saline group (*p* < 0.001). There was also a clinically significant difference in burning or stinging score and cCFS (*p* < 0.01). The ocular AEs were of mild severity, and no severe AEs were reported [26].

The randomized controlled trial validity assessment for perfluorohexyloctane, MOJAVE Study, is summarized in Table 5.

### 3.5. Varenicline

Another new agent, varenicline nasal spray (Tyrvaya), gained FDA approval in 2021 [27]. Varenicline works by binding to cholinergic receptors present in the nasal mucosa. This activates the trigeminal parasympathetic pathway, leading to a spike in tear production [28,29]. Varenicline is administered twice daily into each nostril [30]. 

In a phase 2b multicenter double-masked, randomized, vehicle-controlled trial (ONSET-1), the efficacy of varenicline solution at different doses was observed. One hundred and eighty-two patients were randomized into one of the three varenicline (OC-01) groups (n = 139) (0.03 mg vs. 0.06 mg vs. 0.006 mg) or the vehicle group (n = 43). The primary endpoint measured a change in STS, and the secondary endpoints involved a change in EDS. To assess safety, the study measured TEAE. Patients who received OC-01 0.03 or 0.06 mg showed a clinically significant improvement in tear film production (*p* < 0.001). Participants receiving OC-01 0.03 mg had a statistically significant reduction in EDS (*p* = 0.021), while those receiving OC-01 0.06 mg did not demonstrate clinical significance. Subjects receiving any OC-01 experienced at ≥1 TEAE (70–93%) compared to the vehicle group (26%). Overall, this study proved that both the OC-01 0.03 and 0.06 mg doses may improve the signs and symptoms of DED. OC-01 products were well-tolerated and had mild side effects, including sneezing and coughing [31].

The randomized controlled trial validity assessment for varenicline, ONSET-1, is summarized in Table 6.

In a phase 3 randomized, double-masked, multicenter, vehicle-controlled trial (ONSET-2), the safety and efficacy of varenicline solution at difference doses (0.06 mg vs. 0.03 mg) were assessed to treat DED. Seven hundred and fifty-eight participants were randomized into one of the two varenicline (OC-01) groups (n = 506) or the vehicle group (n = 252). The primary endpoint measured STS changes ≥ 10 mm, while the secondary endpoint was changes in both STS and EDS in a controlled adverse environment (CAE) chamber. The study also measured TEAE (treatment-emergent adverse events) to assess safety. The results showed that a statistically significant percentage of participants achieved 10 mm or more changes in STS in the OC-01 0.03 mg and 0.06 mg groups, with 47.3% and 49.2% of subjects, respectively. The difference for EDS was not clinically significant, and 86.5% of subjects reported ≥ 1 TEAE, including throat irritation, cough, sneezing, and instillation site irritation. This clinical trial demonstrated that varenicline solution at both doses was well-tolerated and significantly reduced the DED symptoms [32].

The randomized controlled trial validity assessment for varenicline, ONSET-2, is summarized in Table 7.

After critically appraising the seven articles, the validity assessment revealed a low risk of bias with minor concerns regarding poor follow-up and moderate drop-out rates (see Table 1, Table 2, Table 3, Table 4, Table 5, Table 6 and Table 7). Overall, all RCTs used in this paper were considered to have high validity.

Table 8 provides a comparison of all five novel DED medications. Having a good understanding of these distinctions, pharmacists can assist in resolving treatment challenges by recommending step-up therapies earlier in the treatment, especially for refractory DED cases.

## 4. Discussion

Non-prescription artificial tears made from ingredients such as carboxymethylcellulose are a common treatment for ADDE. These products provide lubrication to the eye surface and relieve symptoms such as dryness and the feeling of a foreign object in the eye. In addition, treatment with carboxymethylcellulose may improve visual acuity and lessen damage to the eye. It is important to note, however, that while carboxymethylcellulose can provide temporary symptomatic relief, it cannot stop the progression of the disease. Furthermore, pharmacists must inform patients that many carboxymethylcellulose multi-dose products contain preservatives that can cause toxic conjunctivitis if used more than four to six times per day. Preservative-free formulations (PF) are recommended when more frequent use is required to minimize the risk of toxic conjunctivitis.

Restasis, also known as cyclosporine 0.05%, has been a commonly prescribed ophthalmic treatment for DED since its approval in 2003. It functions as an immunosuppressive agent, actively reducing inflammation and increasing tear production. While it has proven to be quite effective, it requires long-term use, and patients may not experience symptom relief until at least six months of consistent use. To address this issue, clinicians can opt for newer treatment options developed to minimize the duration of use and the intensity of DED symptoms [14].

Cyclosporine 0.09% presents a significant clinical improvement compared to Cyclosporine 0.05%, which came to the market more than twenty years ago. The aqueous nanomicelle formulation of cyclosporine 0.09% prevents its degradation, leading to higher bioavailability and efficacy in ocular tissue [19]. Clinical trials have demonstrated that cyclosporine 0.09% can improve corneal staining, conjunctival staining, and STS. Despite these benefits, cyclosporine 0.09% is a new medication that only gained approval in 2018, and many insurance companies mandate a trial of both lifitegrast and cyclosporine 0.05% before approving cyclosporine 0.09%. Additionally, the efficacy of cyclosporine 0.09% has yet to be evaluated in patients with severe intensity DED.

Loteprednol etabonate is a highly promising medication due to its exceptional lipophilicity, which allows it to penetrate cell membranes easily and reach the intended sites of action. It also boasts a high therapeutic index, effective at low doses with minimal side effects. In addition, the drug exhibits significant binding to corticosteroid–glucocorticoid receptors, which are important targets for anti-inflammatory and immunosuppressive therapies. Overall, loteprednol etabonate holds excellent potential for treating patients refractory to cyclosporine 0.05% therapy.

Perfluorohexyloctane is a groundbreaking ophthalmic medication recently approved to tackle tear evaporation directly. This medication is considered an ideal option for patients who suffer from EDE, a condition resulting from the oil deficiency in the tear film layer, increasing tear evaporation. Perfluorohexyloctane targets this underlying cause, thus reducing symptoms such as eye redness, burning, and irritation. Perfluorohexyloctane is a safer option for patients with preservative sensitivity because it contains no preservatives. Preservatives can cause further ocular tissue damage and irritation in some patients, and some individuals may have an allergic reaction to them. In summary, perfluorohexyloctane has emerged as a promising medication for treating evaporative dry eye syndrome. Its direct targeting of tear evaporation and lack of preservatives make it a safe and practical option for patients who suffer from this condition.

Varenicline uses a more convenient administration route that offers several benefits to patients. Unlike most eye medications that treat DED, varenicline does not cause ocular discomfort. This is because it is administered nasally and does not come into direct contact with the eyes. Additionally, varenicline is free from preservatives, which further minimizes the risk of toxic conjunctivitis. The nasal administration of varenicline also makes it an ideal option for patients who wear contact lenses. Unlike other DED medications, there is no need to remove contact lenses before using varenicline. This saves time and hassle, making it a more convenient option for contact lens wearers. Furthermore, varenicline is particularly well-suited for patients with certain health conditions. For example, it is an excellent choice for patients with glaucoma, arthritis, or Parkinson’s disease. Since varenicline is administered nasally, it does not cause the same joint pain or tremors that some eye drops can trigger in these patients. Overall, varenicline is a safe and effective choice for patients seeking relief from DED symptoms.

It is worth noting that even though varenicline is a widely used medication for smoking cessation, its nasal spray form is not effective due to its minimal absorption into the bloodstream [33]. Moreover, each dose of varenicline nasal spray delivers only 0.03 mg of varenicline, significantly lower than the starting dose of 0.5 mg required for smoking cessation [34]. Additionally, no evidence supports the simultaneous usage of varenicline nasal spray with cyclosporine, lifitegrast, or loteprednol [35]. However, one study has suggested that varenicline nasal spray may induce the production of more natural tears than lifitegrast, indicating its potential benefits [36].

Regarding pharmacy practice application and therapy roles, artificial tears will likely remain the first-line therapy for mild DED. For a step-up therapy, clinicians may want to consider the new cyclosporine 0.09% instead of the 0.05% version. For those who do not respond to cyclosporine, Loteprednol etabonate is another option. Since many DED patients also have an evaporative disease component, perfluorohexyloctane can be a good choice for them. Finally, Varenicline offers a new drug delivery route for patients who could benefit from nasal medication, such as contact lens users.

## 5. Limitations

Our study has certain limitations that need to be acknowledged. Firstly, we could not find any head-to-head trials that compare the effectiveness of these novel eye medications. It is also uncertain whether using these medications together provides synergistic benefits. It should also be noted that our research only focuses on the manufacturers’ landmark clinical trials that resulted in the FDA approval of these drugs. Lastly, the FDA has only recently approved perfluorohexyloctane, and more post-marketing surveillance must be conducted to monitor its efficacy.

## 6. Conclusions

Dry eye disease (DED) is a prevalent condition, particularly among women and older individuals. Typically, ophthalmic agents like carboxymethylcellulose and cyclosporine 0.05% have been used by clinicians to alleviate dryness and increase tear production. Unfortunately, these treatments may contain preservatives or require prolonged usage before patients experience any relief. Additionally, not all patients respond favorably to these therapies. 

We are now able to manage dry eye disease with a new generation of FDA-approved medications, including Lifitegrast 5%, cyclosporine 0.09%, varenicline nasal spray, loteprednol etabonate 0.25%, and perfluorohexyloctane, that utilize unique mechanisms and nanotechnology. Our randomized controlled trial validity comparison found the manufacturers’ trials to be robust with predominantly low bias. Our review found that these five novel medications demonstrated promising efficacy and minimal adverse effects. Cyclosporine and loteprednol are effective when artificial tears prove insufficient, while perfluorohexyloctane reduces tear film evaporation and is preservative-free. Varenicline offers drug delivery through the nasal route and is ideal for contact lens users. In summary, these five new medications offer alternative treatment options for DED patients whose condition is not managed by current products and may provide additional benefits with minimal side effects.

## 7. Future Directions

New medications for dry eye disease, including cyclosporine 0.09%, loteprednol etabonate 0.25%, lifitegrast 5%, varenicline nasal spray, and perfluorohexyloctane, have been shown to be both safe and effective. While mild to moderate adverse effects are possible, the benefits of these medications outweigh the risks. Further evaluation through post-marketing surveillance is essential to assess the long-term safety and use of these medications in special populations such as pregnant women, lactating mothers, and children. It would also be helpful to conduct head-to-head trials to compare the side effects and efficacy profiles of these medications and determine the best first-line therapy, as well as potential synergistic benefits with combination therapy.

## Figures and Tables

**Figure 1 pharmacy-12-00019-f001:**
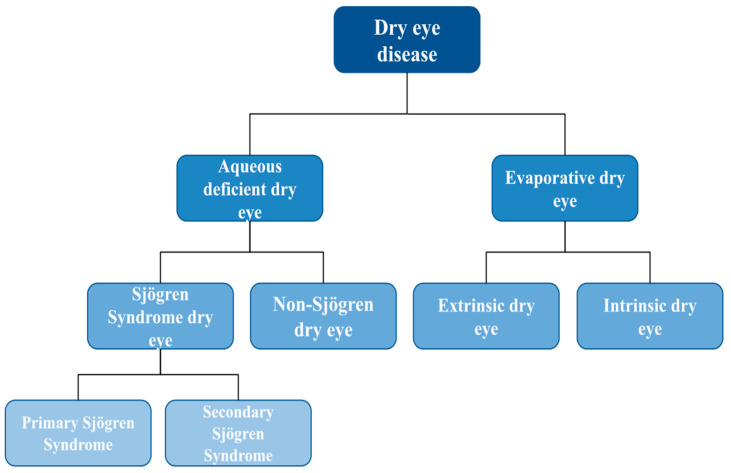
DEWS II classification of DED.

**Figure 2 pharmacy-12-00019-f002:**
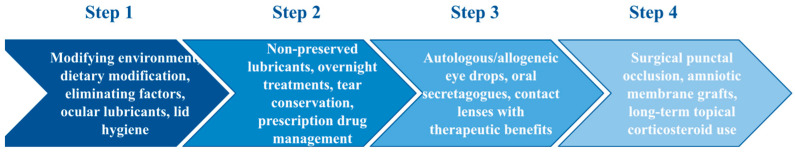
The 2018 AAO treatment algorithm for DED.

**Figure 3 pharmacy-12-00019-f003:**
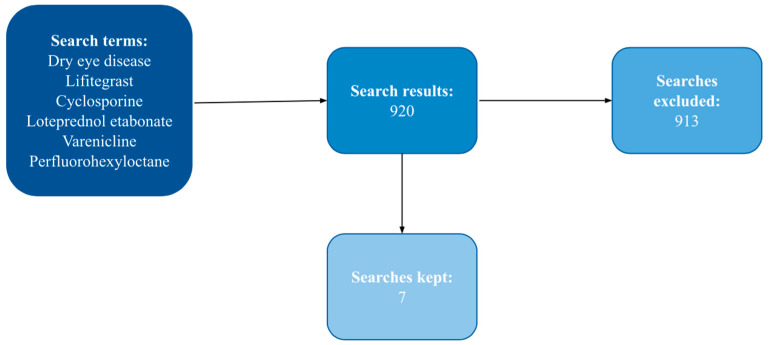
Literature search and article inclusion/exclusion.

**Table 1 pharmacy-12-00019-t001:** Lifitegrast Ophthalmic Solution 5.0% RCT validity table.

		Risk of Bias		
Lifitegrast Ophthalmic Solution 5.0% [17]		Low	Moderate	High
F	There was no mention of a follow-up; 5.7% of participants discontinued the study.		X	
R	Subjects were randomized 1:1 facilitated by an interactive web response system.	X		
I	ITT population included all randomized subjects.	X		
S	All subjects had similar baseline characteristics.	X		
B	The study was double-masked.	X		
E	There was no mention of contamination.	X		

**Table 2 pharmacy-12-00019-t002:** Cyclosporine ophthalmic solution 0.09% RCT validity table.

		Risk of Bias		
Cyclosporine Ophthalmic Solution 0.09% [19]		Low	Moderate	High
F	Follow-up visits were conducted on days 28, 56, and 84.	X		
R	Subjects were randomized via an interactive web response system.	X		
I	ITT population included all randomized subjects.	X		
S	All subjects had similar baseline characteristics.	X		
B	The study was double-masked. Patients, investigators, clinical site staff, and monitoring personnel remained masked.	X		
E	There was no mention of contamination.	X		

**Table 3 pharmacy-12-00019-t003:** Loteprednol etabonate ophthalmic suspension 0.25% RCT validity table.

		Risk of Bias		
Loteprednol Etabonate Ophthalmic Suspension 0.25% [21]		Low	Moderate	High
F	There was no mention of a follow-up; 4/2868 (0.1%) participants were lost to follow up.		X	
R	Subjects were randomized in a 1:1 ratio.	X		
I	The study design considered the ITT population.	X		
S	All subjects had similar baseline characteristics.	X		
B	The study was double-masked.	X		
E	There was no mention of contamination.	X		

**Table 4 pharmacy-12-00019-t004:** Perfluorohexyloctane RCT, GOBI Study, validity table.

		Risk of Bias		
Perflurohexyloctane (GOBI Study) [25]		Low	Moderate	High
F	Follow-up visits carried out at weeks 2, 4, and 8.	X		
R	Subjects were randomized in a 1:1 ratio (interactive web response system).	X		
I	Data were analyzed in the per-protocol population.		X	
S	All subjects had similar baseline characteristics.	X		
B	The study was double-masked.	X		
E	There was no mention of contamination.	X		

**Table 5 pharmacy-12-00019-t005:** Perfluorohexyloctane RCT validity table.

		Risk of Bias		
Perflurohexyloctane (MOJAVE Study) [26]		Low	Moderate	High
F	Follow-up visits carried out at weeks 2, 4, and 8.	X		
R	Subjects were randomized in a 1:1 ratio (interactive web response system).	X		
I	Data was analyzed in the per-protocol population.	X		
S	All subjects had similar baseline characteristics.	X		
B	The study was double-masked.	X		
E	There was no mention of contamination.	X		

**Table 6 pharmacy-12-00019-t006:** Varenicline solution nasal spray RCT, ONSET-1, validity table.

		Risk of Bias		
Varenicline Solution Nasal Spray [31]		Low	Moderate	High
F	There was no mention of a follow-up; 5 participants withdrew from the study.	X		
R	Subjects were randomized 1:1:1:1.	X		
I	The statistical analysis was performed using the ITT population.	X		
S	All subjects had similar baseline characteristics.	X		
B	The study was double-masked.	X		
E	There was no mention of contamination.	X		

**Table 7 pharmacy-12-00019-t007:** Varenicline solution nasal spray RCT, ONSET-2, validity table.

		Risk of Bias		
Varenicline Solution Nasal Spray [32]		Low	Moderate	High
F	There was no mention of a follow-up.		X	
R	Subjects were randomized 1:1:1.	X		
I	ITT population included all randomized subjects.	X		
S	All subjects had similar baseline characteristics.	X		
B	The study was double-masked.	X		
E	There was no mention of contamination.	X		

**Table 8 pharmacy-12-00019-t008:** Comparison of cyclosporine, loteprednol, lifitegrast, varenicline nasal spray, and perfluorohexyloctane.

	Cyclosporine Ophthalmic Solution 0.09%	Loteprednol Etabonate Ophthalmic Suspension 0.25%	Lifitegrast Ophthalmic Solution 5.0%	Varenicline Solution Nasal Spray	Perfluorohexyloctane
Dosage and administration	One drop twice daily into each eye	Shake for two to three seconds before use. Instill one to two drops into each eye four times daily	One drop twice daily into each eye using a single-use container	One spray into each nostril twice daily (12 h apart)	Instill one drop four times daily into each eye
Adverse reactions	Pain on instillation of drops (22%), conjunctival hyperemia (6%), blepharitis, eye irritation, headache, and urinary tract infection (1–5%)	Elevated intraocular pressure	Instillation site irritation, dysgeusia, and reduced visual acuity (5–25%), blurred vision, conjunctival hyperemia, eye irritation, headache, increased lacrimation, eye discharge, eye discomfort, pruritus, and sinusitis (1–5%)	Sneezing (82%), cough (16%), throat irritation (13%), instillation-site (nose) irritation (8%)	Blurred vision, conjunctival redness (1–3%)
Special populations	Pregnancy: No available data Lactation: No available data Pediatrics: Safety and efficacy have not been established Geriatrics: No differences in safety or efficacy observed				
Storage and handling	Store at 20–25 °C (68–77 °F) Store single-use vials in the original foil pouch	Store upright at 15–25 °C (59–77 °F) Do not freeze After opening, use until the expiration date	Store at 20–25 °C (68–77 °F)	Store at 20–25 °C (68–77 °F) Do not freeze Discard nasal spray bottle 30 days after opening	Store at 20–25 °C (68–77 °F) After opening, may be used until the expiration date

## Data Availability

Not applicable.

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
