# Peer review of "Managing Dry Eye Disease with Novel Medications: Mechanism, Study Validity, Safety, Efficacy, and Practical Application"

_pharmacy, 2024, doi:10.3390/pharmacy12010019_

Round 1

Reviewer 1 Report

Comments and Suggestions for Authors

The authors have reviewed five new medications for dry eyes, namely: lifitegrast, cyclosporine, loteprednol etabonate, varenicline and perflurohexyloctane. The authors have produced nicely organised tables but it seems to be too much information and it is difficult for readers to follow the writing. For example, the introduction (section 1) and background (section 4) should be combined. Line 127 belongs in the discussion section. Lines 61-64 belong to the methods section. The tables should be presented in the results section and it is not clear what are the information has been extracted from their literature search. In terms of the methods, it is puzzling that the authors conducted a “comprehensive search for “landmark trials that led to the FDA approval” of these drugs. If the authors are only interested in the papers that are of importance to the approval of the drugs, why do they need to conduct a comprehensive search? The term “landmark” implies that these trials are so important that it can’t be missed, so it does not make sense to conduct a “comprehensive search”. For the title, does the author mean clinical applications or practical applications? The authors should also try to quote the original papers, e.g. references #2 to #4. For example, Dalton did not report that the rise of DED is due to the increased use of soft contact lenses.

Author Response

Dear reviewer,

We sincerely appreciate your valuable feedback. We have modified the manuscript according to your suggestions and provided our responses as follows.

The authors have produced nicely organised tables but it seems to be too much information and it is difficult for readers to follow the writing.

  • The information has been reduced significantly.
  • Original tables 1,3,5,7, and 10 were removed because some of this information was already discussed in the paper and the rest was summarized in the new table 8.

For example, the introduction (section 1) and background (section 4) should be combined.

  • Both sections have been combined into one introduction section.

Line 127 belongs in the discussion section.

  • Line 127 (Pharmacists should only recommend preservative-free (PF) ophthalmics for patients with sensitivity to preservatives.) has been moved into the discussion section.

Lines 61-64 belong to the methods section.

  • Lines 61-64 (To evalutate the validity of the seven randomized control trials (RCT), a critical appraisal tool (FRISBE mnemonic) was used. The mnemonic stands for “follow up”, “randomization”, “intent-to-treat analysis”, similar baseline characteristics”, “blinding”, and “equal treatment”. After critically appraising the seven articles, the results revealed a low risk of bias with minor concerns regarding poor follow-up and moderate drop-out rates. Overall, all RCTs used in this paper were considered to have high validity.) has been moved to the methods section.

The tables should be presented in the results section and it is not clear what are the information has been extracted from their literature search.

  • The tables have been moved to the results section. The RCT validity tables have citations to indicate those are the information from the literature search.

In terms of the methods, it is puzzling that the authors conducted a “comprehensive search for “landmark trials that led to the FDA approval” of these drugs. If the authors are only interested in the papers that are of importance to the approval of the drugs, why do they need to conduct a comprehensive search? The term “landmark” implies that these trials are so important that it can’t be missed, so it does not make sense to conduct a “comprehensive search”.

  • The term “Comprehensive” to describe the search was removed.

For the title, does the author mean clinical applications or practical applications?

  • Practice; meaning what roles of therapy do these novel medications have. How we are going to apply what we learn from these clinical trials in pharmacy practice

The authors should also try to quote the original papers, e.g. references #2 to #4. For example, Dalton did not report that the rise of DED is due to the increased use of soft contact lenses.

  • Has been corrected.

Thank you again for taking the time to review. We appreciate it. Happy and healthy 2024!

Reviewer 2 Report

Comments and Suggestions for Authors

Dear Authors,

Thank you for submitting your article for publication in "Pharmacy" Journal.

This review article presents a promising overview of recent advancements in dry eye disease management, specifically focusing on five recently approved treatments. Although the content holds potential, significant revisions are necessary to improve clarity, structure, and adherence to publication guidelines. The areas which need improvement are as follow:

  • Title: The title needs revising to more accurately reflect the article's scope and focus.
  • Abstract: The abstract is too long and lacks a structured format. Consider revising it to a concise, structured abstract (e.g., IMRaD) that highlights the key points, including objectives, methods, findings, and conclusions.
  • The article's objectives lack focus. Clarify the review's purpose and specific assessment criteria.
  • Manuscript length: The manuscript exceeds the recommended word count. Condense the content by streamlining sections, combining redundant information, and focusing on the most relevant details.
  • Structure: The manuscript deviates from the recommended IMRaD structure. Combine the introduction and background into a single section, clearly distinguish methods from results, and move literature findings from the discussion into the results section.
  • Results/Discussion: Enhance the results section by incorporating information currently buried in the discussion. Conversely, strengthen the discussion by providing expert interpretation and evaluation of the findings, linking them to existing literature.
  • Conclusion: Rewrite the conclusion to summarize the key findings and offer clear recommendations for practice and future research.

Recommendations:

  • Revise the title to accurately reflect the article's focus.
  • Rewrite the abstract in a structured format.
  • Condense the manuscript to meet word count guidelines.
  • Reorganize the manuscript to follow the IMRaD structure.
  • Strengthen the results and discussion sections by integrating related information.
  • Rewrite the conclusion to summarize key findings and offer clear recommendations.

Thank you

The reviewer 

Comments on the Quality of English Language

The review article is very interesting, holds promise, and has the potential to be a valuable contribution to the field. However, significant revisions are necessary to address weaknesses and enhance the article's clarity, coherence, and impact, as detailed in the authors section.

Author Response

Dear reviewer,

We sincerely appreciate your valuable feedback. We have modified the manuscript according to your suggestions and provided our responses as follows.

This review article presents a promising overview of recent advancements in dry eye disease management, specifically focusing on five recently approved treatments. Although the content holds potential, significant revisions are necessary to improve clarity, structure, and adherence to publication guidelines. The areas which need improvement are as follow:

  • Title: The title needs revising to more accurately reflect the article's scope and focus.
    • Excellent point. The title has been revised to: “Managing Dry Eye Disease with Novel Medications: Mechanism, Study Validity, Safety, Efficacy, and Practice Application”
  • Abstract: The abstract is too long and lacks a structured format. Consider revising it to a concise, structured abstract (e.g., IMRaD) that highlights the key points, including objectives, methods, findings, and conclusions.

The format of the abstract has been revised to follow the “IMRaD” model. Please see the following abstract:

  • Abstract: Dry Eye Disease (DED) is a common condition that affects mainly older individuals and women. It is characterized by reduced tear production and increased tear evaporation. Symptoms include burning, irritation, tearing, and blurry vision. This paper reviews key trials of various new DED treatments, including their mechanism of action, study outcomes, safety, and efficacy. The paper also includes a critical assessment of the trial's validity and potential pharmacy applications of these new treatments. The literature search was conducted through PubMed, Cochrane Central Register of Controlled Trials, and Google Scholar. Keywords: “Dry Eye Disease”, “lifitegrast”, “cyclosporine”, “loteprednol etabonate”, “varenicline nasal spray”, and “perfluorohexyloctane” were used to identify these medications’ landmark trials. The articles deemed these medications safe and efficacious, with minimal side effects. Our randomized controlled trial validity comparison found the trials robust with predominantly low bias. Cyclosporine and loteprednol are effective when artificial tears fail, while perfluorohexyloctane reduces tear film evaporation and is preservative-free. Varenicline offers drug delivery via the nasal route and is appropriate for contact lens users. In conclusion, these FDA-approved novel medications exhibit safety and efficacy in managing DED. Further research is needed on long-term outcomes, efficacy and side effects comparisons, and combination therapy benefits.
  • Introduction: Dry Eye Disease (DED) is a common condition that affects mainly older individuals and women. It is characterized by reduced tear production and increased tear evaporation. Symptoms include burning, irritation, tearing, and blurry vision.
  • Objective: This paper reviews key trials of various new DED treatments, including their mechanism of action, study outcomes, safety, and efficacy. The paper also includes a critical assessment of the trial's validity and potential pharmacy applications of these new treatments.
  • Methods: The literature search was conducted through PubMed, Cochrane Central Register of Controlled Trials, and Google Scholar. Keywords: “Dry Eye Disease”, “lifitegrast”, “cyclosporine”, “loteprednol etabonate”, “varenicline nasal spray”, and “perfluorohexyloctane” were used to identify these medications’ landmark trials.
  • Findings: The articles deemed these medications safe and efficacious, with minimal side effects. Our randomized controlled trial validity comparison found the trials robust with predominantly low bias. Cyclosporine and loteprednol are effective when artificial tears fail, while perfluorohexyloctane reduces tear film evaporation and is preservative-free. Varenicline offers drug delivery via the nasal route and is appropriate for contact lens users.
  • Conclusions: In conclusion, these FDA-approved novel medications exhibit safety and efficacy in managing DED. Further research is needed on long-term outcomes, efficacy and side effects comparisons, and combination therapy benefits.

  • Objectives: lack focus. Clarify the review's purpose and specific assessment criteria.
    • Objective: This paper reviews key trials of various new DED treatments, including their mechanism of action, study outcomes, safety, and efficacy. The paper also includes a critical assessment of the trial's validity and potential pharmacy applications of these new treatments.
  • Manuscript length: The manuscript exceeds the recommended word count. Condense the content by streamlining sections, combining redundant information, and focusing on the most relevant details.
    • The manuscript has been revised; information has been combine or condense to focus on the objectives
  • Structure: The manuscript deviates from the recommended IMRaD structure. Combine the introduction and background into a single section, clearly distinguish methods from results, and move literature findings from the discussion into the results section.
  • Introduction and background have been combined into “Introduction”. Methods and results have been separated – results section now focuses on the clinical trials.
  • Results/Discussion: Enhance the results section by incorporating information currently buried in the discussion. Conversely, strengthen the discussion by providing expert interpretation and evaluation of the findings, linking them to existing literature.
    • Some discussion materials that should have been in the results section has been moved to the discussion section. The discussion has been strengthen to enhance expert interpretation, findings evaluation and roles of therapy.
  • Conclusion: Rewrite the conclusion to summarize the key findings and offer clear recommendations for practice and future research.
    • Conclusion has been rewritten to summarize findings and lessons learned. There is a future directions section after the summary so it’s more clear what future research should direct.

Recommendations:

  • Revise the title to accurately reflect the article's focus.
  • Rewrite the abstract in a structured format.
  • Condense the manuscript to meet word count guidelines.
  • Reorganize the manuscript to follow the IMRaD structure.
  • Strengthen the results and discussion sections by integrating related information.
  • Rewrite the conclusion to summarize key findings and offer clear recommendations.

Thank you

The reviewer 

The review article is very interesting, holds promise, and has the potential to be a valuable contribution to the field. However, significant revisions are necessary to address weaknesses and enhance the article's clarity, coherence, and impact, as detailed in the authors section.

Thank you again for taking the time to review. We appreciate it. Happy and healthy 2024!

Round 2

Reviewer 1 Report

Comments and Suggestions for Authors

The authors have done a well written review of five new medications for dry eyes, namely: lifitegrast, cyclosporine, loteprednol etabonate, varenicline and perflurohexyloctane. corrections are satisfactory. They reported a low risk of bias in the RCTs. have provided a neatly summarized comparative table for these medications, which can be helpful for pharmacists to use as a guide.

Reviewer 2 Report

Comments and Suggestions for Authors

Dear Authors,

I'm pleased to see the significant improvements you've made to the manuscript in response to my comments and suggestions. The flow is noticeably smoother, the text is more comprehensible, and the scientific rigor is demonstrably stronger. The revised copy has addressed my key concerns and enhanced the overall clarity and impact of the work.

I wish you the very best with your publication efforts. I feel confident that this revised manuscript will be well-received by the journal and the broader scientific community.

Best regards,

The Reviewer